# In Vitro Characterization of *chIFITMs* of Aseel and Kadaknath Chicken Breeds against Newcastle Disease Virus Infection

**DOI:** 10.3390/biology12070919

**Published:** 2023-06-27

**Authors:** Muthusamy Malarmathi, Nagarajan Murali, Mani Selvaraju, Karuppusamy Sivakumar, Vasudevan Gowthaman, Vadivel Balasubramanian Raghavendran, Angamuthu Raja, Sunday O. Peters, Aranganoor Kannan Thiruvenkadan

**Affiliations:** 1Veterinary College and Research Institute, Tamil Nadu Veterinary and Animal Sciences University, Namakkal 637 002, India; murali.n@tanuvas.ac.in (N.M.); selvaraju.m@tanuvas.ac.in (M.S.); raghavendran@tnau.ac.in (V.B.R.); raja.a@tanuvas.ac.in (A.R.); 2Faculty of Food and Agriculture, The University of the West Indies, St Augustine 999183, Trinidad and Tobago; sivakumar.karuppusamy@sta.uwi.edu; 3Poultry Disease Diagnosis and Surveillance Laboratory, Tamil Nadu Veterinary and Animal Sciences University, Namakkal 637 002, India; gowthaman.v@tanuvas.ac.in; 4Department of Animal Science, Berry College, Mount Berry, GA 30149, USA; speters@berry.edu; 5Veterinary College and Research Institute, Tamil Nadu Veterinary and Animal Sciences University, Salem 636 112, India

**Keywords:** *ISG*, *IFITM* gene, *Mx*, interferon, CEF, Aseel, Kadaknath, Newcastle disease, viral load

## Abstract

**Simple Summary:**

Aseel and Kadaknath are indigenous chicken breeds in India. Due to their distinctiveness, these two breeds are becoming more important. Aseel has a reputation for cockfighting and having high-quality meat. The black flesh of Kadaknath is famous, and it is mostly raised for its meat and eggs. Both breeds have heat and disease tolerances. The chicken interferon-inducible transmembrane protein (*chIFITM*) genes function by preventing the entry of viruses into host cells, inhibiting viral replication, and controlling the viral load. This research was conducted to ascertain the level of *chIFITM* gene expression against the Newcastle disease virus (NDV) in chicken embryo fibroblast cells of both breeds. There are five members of the *chIFITM* gene family: *chIFITM*1, *chIFITM*2, *chIFITM*3, *chIFITM*5, and *chIFITM*10. *chIFITM*1, 2, and 3 have immune-related activity; as a result, these three are referred to as immune-related *IFITM* (*IR*-*IFITM*). All the *chIFITM* genes were shown to be highly expressed in the CEF cells of both breeds during our investigation. The Kadaknath CEF cells had a significantly low viral load and a high quantity of mRNAs for the *chIFITM* genes when the breeds were compared. Aseel cells demonstrated an earlier onset of NDV-induced cytopathic changes compared to Kadaknath.

**Abstract:**

Newcastle disease (ND) is highly contagious and usually causes severe illness that affects *Aves* all over the world, including domestic poultry. Depending on the virus’s virulence, it can impact the nervous, respiratory, and digestive systems and cause up to 100% mortality. The *chIFITM* genes are activated in response to viral infection. The current study was conducted to quantify the mRNA of *chIFITM* genes in vitro in response to ND viral infection. It also examined its ability to inhibit ND virus replication in chicken embryo fibroblast (CEF) cells of the Aseel and Kadaknath breeds. Results from the study showed that the expression of all *chIFITM* genes was significantly upregulated throughout the period in the infected CEF cells of both breeds compared to uninfected CEF cells. In CEF cells of the Kadaknath breed, elevated levels of expression of the *chIFITM*3 gene dramatically reduced ND viral growth, and the viral load was 60% lower than in CEF cells of the Aseel breed. The expression level of the *chIFITMs* in Kadaknath ranged from 2.39 to 11.68 log_2_ folds higher than that of control CEFs and was consistently (*p* < 0.01) higher than Aseel CEFs. Similar to this, the*IFN-γ* gene expresses strongly quickly and peaks at 13.9 log_2_ fold at 48 hpi. Based on these cellular experiments, the Kadaknath breed exhibits the potential for greater disease tolerance than Aseel. However, to gain a comprehensive understanding of disease resistance mechanisms in chickens, further research involving in vivo investigations is crucial.

## 1. Introduction

Newcastle disease (ND) has been globally distributed, and the causative agent, Newcastle disease virus (NDV), belongs to the *Paramyxoviridae* family, genus *Avulavirus,* and is designated as *AvianParamyxovirus-1* (*APMV*-*1*). NDV is a non-segmented, negative-sense, single-stranded enveloped RNA virus with an approximately 15kb genome that encodes hemagglutinin-neuraminidase (HN), nucleoprotein (NP), fusion (F), phosphoprotein (P), matrix (M), and RNA-dependent RNA polymerase (L) [1]. The ND virus has a wide range of hosts, and infection has been reported in 250 *Avianspecies* in the world by either natural or experimental mechanisms [2]. Depending on the viral pathotype, the incidence of ND that affects poultry manifests as gastrointestinal, respiratory, and neurological conditions that can result in up to 100% mortality [3,4]. ND has a high impact on the poultry industry through heavy economic loss aroused due to heavy mortality and production loss and also due to extensive attention paid to the prevention and treatment of this disease, such as standard vaccination protocols and biosecurity measures [5]. It was calculated that thirteen layer farms in the Gujarat state of India suffered a total economic loss of $4588(₹ 37,19,223) per year [6]. Various disease prevention strategies and vaccinations are not effective due to the complex genetic diversity of viruses. So, it is necessary to develop a chicken population that is naturally disease-resistant.

During viral infection, *Type I interferon* defense mechanisms are triggered to express a set of genes against viral infections known as interferon-stimulated genes (*ISGs*) [7,8]. The *IFITM* (interferon-inducible transmembrane) gene is one of these *ISGs* and has been shown to prevent the propagation of several highly virulent viral pathogens, such as the coronavirus responsible for the severe acute respiratory syndrome (*SARS*), the filoviruses Marburg and Ebola, the influenza A viruses (*IAVs*), and flaviviruses (dengue virus) [9,10,11]. Scientists discovered that *chIFITM* gene expression has a negative correlation with the emergence of the influenza virus and its titre in vitro, indicating that *chIFITMs* 1,2, and 3 have a functional role in the management of viral infections [12,13]. In chickens, *IFITM* genes were located in Chromosome 5 and found in two loci, one containing the various numbers of immune-related (*IR*)-*IFITM* (*IFITM1, 2,* and *3*) genes, and the *IFITM*5 gene and the *IFITM10* gene in another locus [14,15]. Lanzet al. found that *swIFITMs (swine IFITMs)* had a dose-dependent restriction against *IAV* after infecting porcine *HEK293*-T cells with *IAV A/WSN/33 (WSN)* for 24 or 48 h [16]. sw*IFITM*2 and -3 were expressed at late endosomes and have the most potent antiviral activity against *IAV* in porcine cells. Furthermore, no sw*IFITM*5 expression was detected in any of the tested cell lines, indicating that *IFITM5* does not have a significant role in immune function [16]. Further, the knockdown of *IFITM*3 in DF-1 cells by siRNA increased the infectivity of a vesicular stomatitis virus G protein pseudo-typed lentiviral vector [17].

Indian chicken breeds often have a long history of adaptation to the local conditions, including the climate, diseases prevalent in the region, and dietary preferences. These factors can contribute to the development of genetic traits that provide them with better immunocompetence, disease resistance, and adaptability to the tropical environment [18]. In a preliminary study conducted at TANUVAS, the immune response of Aseel, Kadaknath, and White Leghorn chickens against sheep RBS cells was investigated. The results of the study indicated that Aseel and Kadaknath chickens demonstrated a higher immune response in comparison to White Leghorn chickens [19]. Therefore, the purpose of this study was to analyze the gene expression pattern of chIFITM between Aseel and Kadaknath chickens, which exhibit high levels of viral resistance. So, we selected the model of chicken embryo fibroblasts (CEFs) to observe chicken *IFITM* gene expression in vitro following infection with NDVs. We compared the expression of *IFN γ* and *Mx* genes in response to NDV infection by quantitative real-time polymerase chain reaction (qRT-PCR).

In addition to providing prospects for a deeper understanding of viral resistance, analysis of these genes in chickens offers potential strategies for preventing viruses in poultry farming. It may be possible to do a selective breeding program in poultry breeds for increased resistance against viral infections. So, it needs to discover the characteristics of resistance and understand how they operate. Further, the observation may have useful implications in terms of vaccine production. Many vaccines are produced in embryonated hen’s eggs or continuous avian cell lines. However, it is well established that the rate-determining step in the manufacture of numerous vaccines is the induction of antiviral immune responses that prevent the replication of vaccine viruses and the high cost involved in maintaining and producing specific pathogen-free (SFP) eggs from chicken.

## 2. Materials and Methods

### 2.1. Ethics Statement

The experiment was conducted with the approval of the Institutional Bio-safety Committee (IBSC) of TANUVAS-Veterinary College and Research Institute, Namakkal, Tamil Nadu, India (Approval Lr. No. 1764/VCRI-NKL/IBSC/2022 dated 11 May 2022 of the Dean, VCRI, Namakkal).

### 2.2. Chicken Embryo Fibroblast Cells

CEF cells were prepared from 9–10-day-old SPF chicken embryos (10 embryos/breed) of Aseel and Kadaknath (Department of Poultry Science, Veterinary College and Research Institute, TNUVAS, Namakkal, India) as previously described [20]. Fibroblastic cells were isolated from the respective embryo by removing the head and viscera, then cut into small pieces using sterile scissors and force. The remaining tissues were washed with PBS and trypsinized for 5 min with 0.25% trypsin and a magnetic stir. Allow pieces to settle, collect supernatant, centrifuge at 1000 rpm for 5 min, and resuspend pellets in growth medium containing Dulbecco’s modified Eagle’s medium (DMEM) (Hi-media, Mumbai, India, Cat. No.AL007S) supplemented with 10% fetal bovine serum (FBS) (Hi-media, Cat. No. RM112-500ML), 1% antibiotics, Antimycotic solution (100×), stabilized (Sigma, St. Louis, MO, USA, Cat.No. A5955) at 37 °C with 5% CO_2_ for 24 h.

### 2.3. Virus

The *velogenic genotype XIII NDV* strain isolated from a field ND outbreak by the Poultry Disease Diagnosis and Surveillance Laboratory, TANUVAS, Namakkal, India, was used in this study. The ND virus was inoculated into the allantoic cavity of 10-day-old SPF-embryonated White Leghorn chicken eggs and incubated at 37 °C for 72 h as per the standard procedures of the Office International Des Epizooties [21]. After 72 h, inoculated eggs were chilled for 30 min, and then allantoic fluid was collected under sterile conditions and stored at −80 °C for further use. ND virus titers were quantified by Haemagglutination assay (HA) and confirmed by PCR amplification of the ND viral *F-gene* using the primer to amplify a 356 bp amplicon (Figure 1) using the primer pair of *NDVF*(5′-GCAGCTGCAGGGATTGTGGT-3′) and *NDVR* (5′-TCTTTGAGCAGGAGGATGTTG-3′) with the cycle condition of initial denaturation 95 °C for 3 min followed by 35 cycles of 94 °C for 45 s, 52 °C for 45 s, 72 °C for 45 s and final extension 72 °C for 5 min [22].

### 2.4. Tissue Culture Infection Doses (TCID_50_)

The viral infective dose was measured by TCID_50_. CEF cells from SPF-embryonated White Leghorn chicken eggs were cultured in 96-well plates and incubated with the cell supernatants of different groups, which had a 10-fold serially diluted viral suspension. Each dilution had five replicates. One h after NDV infection, the supernatants were replaced with DMEM containing 2% fetal bovine serum (FBS). Then it was incubated at 37 °C, observed daily for CPE scoring, and continued scoring daily till the control wells started dying. The TCID_50_ value was determined using Spearman-Karber’s method [23].

### 2.5. Viral Infection

CEFs prepared from Aseel and Kadaknath SPF chicken embryos were seeded 24 h prior to infection in a 25 cm^2^ Tissue culture Flask (T25) (Hi-media, India) at a cell density of approximately 7 × 10^5^ cells/flask. The CEF cells in triplicate were subjected to infection with *velogenic genotype XIII* of Newcastle Disease Virus (NDV) strains that were circulating in Tamil Nadu, South India [24]. The ND viral suspension was diluted to a 50% tissue culture infective dose (TCID_50_) of 10^6^/mL, and 0.5 mL of viral suspension (TCID_50_) was added into the flask, which allowed for viral adsorption by incubating at 37 °C in a humidified atmosphere containing 5% CO_2_ for 1 hr. Afterwards, the growth medium was replaced with DMEM supplemented with 2% FBS. Uninfected cells were regarded as control samples. CEF cells were harvested from uninfected control and infected CEF cells at 3, 6, 12, 24, and 48 h post-infection (hpi) and stored at −80 °C for RNA extraction. Virus load in CEF cells was quantified by an absolute quantification method [25] and gene expression by the relative quantification 2^−ΔΔCt^ method [26].

### 2.6. RNA Extraction and cDNA Synthesis

Total RNA was extracted from the infected and uninfected control group CEF cells of both breeds at each time point by the Trizol method using RNAiso Plus, Takara (Cat. # 9109) (Total RNA extraction reagent). RNA in each sample was quantified using Thermo Scientific’s Nanodrop spectrophotometer (Thermo Scientific, Waltham, MA, USA). Approximately 1 µg of RNA from each sample was used for complementary DNA (cDNA) synthesis by using the iScript cDNA Synthesis Kit (Bio-Rad, Hercules, CA, USA, Cat # 1708891) according to the manufacturer’s protocol, which follows a method of Reverse Transcription (RT) with a random primer.

### 2.7. Primer Pair Design

RT-qPCR primers for *chIFITMs* genes were designed using Primer-BlAST with a length of 20 to 23 bases and amplicon sizes ranging from 115 to 196 bp. The sequences of these genes were obtained from NCBI (https://www.ncbi.nlm.nih.gov, accessed on 22 April 2021. The primer specificity of each gene was verified using 2.5% agarose gel electrophoresis and melting curve analysis. To validate the specificity of each primer pair, it was verified by in silico PCR with the NCBI sequence database using NCBI PRIMER BLAST.

### 2.8. Quantitative Real-Time Polymerase Chain Reaction (qRT-PCR)

A total of 4 targeted genes (*chIFITM*1, 2, 3, and 5) expression patterns in ND virus-infected and uninfected control CEF cells of both breeds were studied alone with 2 Positive immune-related genes (*IFN γ* and *Mx*) and 1 housekeeping gene (*β Actin*) (Table 1). The relative expression of specific gene mRNA was quantified, and the absolute quantification of viral load was done by a real-time thermal cycler (IIIumina Real-time machine, San Diego, CA, USA). All reactions were performed in a nuclease-free 48-well qRT-PCR Illumina plate with sealer. The qRT-PCR response was done with a final volume of 20 µL using 10 µL of SYBR Green PCR Master Mix Kit (Bio-rad), 10 pmol of each forward and reverse primer, and 1 µL of cDNA. The cycle condition of qRT-PCR was initial denaturation at 95 °C for 10 min, followed by 40 cycles of 95 °C for 10 s, 60 °C (β Actin, *IFN γ, chIFITM*1, and *Mx* genes), 62 °C (*chIFITM*2, 3, and 5) for 45 s, and 72 °C for 15 s. Further, the target and β-actin genes’ respective cycle threshold (Ct) values were computed. Using the 2^−ΔΔCt^ approach, the relative fold change of the target genes in the infected groups was calculated using the delta Ct of the uninfected control group [26].

### 2.9. Standard Curve Analysis for Detection of Viral Load

A standard curve was constructed by the linear regression method to do absolute quantification of viral load in the samples [25]. A ten-fold serially diluted T-NP plasmid (20 ng/µL, A260/280 ratio = 1.80) with known concentration was used as a standard to construct a standard curve and to obtain a linear regression equation. A standard curve was then generated by plotting *Cq* value against the logarithm of the plasmid copy numbers in the standard. A correlation between NDV-specific nucleoprotein (NP) gene copy numbers and *Cq* values as found by using Pla-rt13 and Pla-rt14 primer pairs specific for the nucleoprotein (NP) region of the NDV genome (Table 1). The qPCR cycle condition was initially 95 °C for 5 min, followed by 40 cycles of 95 °C for 15 s, 60 °C for 30 s, and finally 72 °C for 30 s. A melting curve analysis was used to determine the specificity of qPCR primers. A partial regression equation was obtained as 𝑌 = −2.25x + 14.09 (Figure 1). R^2^ value ranging from 0.994 to 0.932 and they indicate strong and linear relationships between the *Cq* value and the number of gene copies in the sample.

### 2.10. Statistical Analysis

The *t*-test and one-way *ANOVA* were conducted to analyze real-time PCR data. The mean (n = 3 per time point/breed for each infected and control) and standard error of the mean are in log_2_(2^−ΔΔCt^) used to express data and log_10_(viral copies) used to express viral load. *R software* version 4.2.1 was used to analyze the data, and a *p*-value < 0.05 was used to determine statistical significance. Further, the R program was also used to create the graphical illustrations of the results.

## 3. Results

### 3.1. NDV Infection-Induced Cytopathic Changes and Viral Load

In the current study, we have determined the IFITM gene expression against the live NDV (velogenic genotype XIII, NDV strains) in the CEF cells. Figure 2 highlights the normal morphology of the CEF cells derived from the 9–10-day-old SPF embryonated chicken eggs. The effect of NDV infection in the chicken embryo fibroblast cells was examined under an inverted microscope for their morphology to determine the cytopathic effect (CPE) at 3, 6, 12, 24, and 48 hpi and compared with uninfected control cells. The cell monolayer was intact and morphologically like the control group until 3 and 6 hpi in Aseel and Kadaknath, respectively. However, at 6 and 12 hpi, morphological changes that are typical of the cytopathic effect (CPE), viz., cell rounding, a fusion of infected cells to form syncytia cells, and detachment of cells from the monolayer followed by cell death, were noticed under the light microscope in both breeds (Figure 2).

Further, NDV-infected CEF cells also showed morphological alterations typical of apoptosis: rounding of the cell and cytoplasm vacuolation were noticed in both breeds. However, cell rounding started at 6 h and progressed, and at 48 h, complete cell detachment was observed in Aseel CEF cells. However, it was delayed in the case of Kadaknath CEFs; cell rounding started at 12 h and progressed, and at 48 h, cell detachment progressed. This revealed that CPE advancement was related to virus load and time in infected cells, in addition to confirming virus infection and its replication in infected cells. 

Viral load was calculated by standard curve analysis using the T-NP plasmid as a standard, and we observed a steady increase in viral load for 12 h and 24 h, respectively, in Aseel and Kadaknath CEF cells (Figure 3). In the case of Aseel CEF, viral load was significantly (*p* < 0.01) higher than Kadaknath at 3 hpi (2.01 log_10_), 6 hpi (2.75 log_10_), and 12 hpi (5.38 log_10_). In contrast, in Kadaknath CEF, the viral load was significantly (*p* < 0.01) lower when compared to Aseel at each point: 3 hpi (1.01 log_10_), 6 hpi (1.59 log_10_), and 12 hpi (2.15 log_10_). However, there is no significant difference found at 24 hpi (4.54 and 4.63 log_10_) and 48 hpi (2.98 and 2.95 log_10_) between Aseel and Kadaknath CEF, respectively. In Aseel, the viral load peaked at 12 h, and the largest load was observed at approximately 5.3 log_10_ viral copies in Aseel, while it was significantly (*p* < 0.001) low at 2.15 log_10_ viral copies as in Kadaknath CEF cells at 12 hpi. Whereas, in Kadaknath, the highest load was recorded at 24 h and found to be 4.63 log_10_ viral copies (Figure 3). In the later hours, the viral production gradually decreased. However, the viral load in the Aseel CEF was still higher than in Kadaknath CEF cells at any time, confirming that Kadaknath cells potentially restrict the multiplication of ND viruses compared to Aseel CEF cells.

### 3.2. Expression Analysis of chIFITM Gene in Newcastle Disease Virus-Infected CEF Cells

The relative expression of chIFITM genes in the control (uninfected) and infected cells was quantified by qRT-PCR. Our results showed that the expression of the chIFITM1, 2, 3, 5, IFN-γ,and Mx genes is time-dependent (Table 2 and Figure 4a,b). After ND viral infection of the CEF cells, the mRNA levels of all four selected genes (chIFITM1, 2, 3, and 5) and positive immune-related genes (IFN-γ and Mx) were gradually increased, reaching a peak at different hours post-infection (hpi) in both breeds. In comparison to the calibrator (uninfected control CEF cells), the relative expression of IFN- γ reaches a peak at 48 hpi at 2.59 log_2_ fold and 13.9 log_2_ fold higher in Aseel and Kadaknath, respectively. The results show that the chIFITMs are expressed at basal levels in CEF cells of both breeds. It also demonstrates that the breeds that were studied exhibit various patterns of expression. Compared to Kadaknath CEF, expression of the chIFITMs gene is lower and more variable in Aseel.

In Aseel, only at 12 hpi was expression of the chIFITM2 gene dramatically increased, whereas chIFITM5 upregulation begins at 6 hpi and continues until 24 hpi. In addition, the highest expression in chIFITM2 and 5 genes was recorded as 3.25 log_2_ fold and 3.82 log_2_ fold, respectively, at 6 hpi and 24 hpi. The chIFITM3 gene was strongly expressed from 3 to 12 hpi, with the maximum level being 3.46 log_2_ fold at 6 hpi. Like chIFITM3, chIFITM1 upregulation starts at 3 hpi, but it continues strongly until 48 hpi and reaches a maximum of 6 log_2_ fold at 6 hpi. Mx and chIFITM3 exhibit comparable patterns of expression, with the Mx gene reaching its highest level of expression at 3 hpi as a 4.18 log_2_ fold increase.

As opposed to Aseel, Kadaknath CEF cells express all chIFITM genes strongly and severalfold significantly higher from 3 to 48 h after ND virus infection. At 6 hpi, the highest expression of chIFITM2 was detected. chIFITM1, 5, and Mx were at 24 hpi, but chIFITM2 and IFN-γ were at 48 hpi. In Kadaknath, the expression level among the chIFITMs ranged from 2.39 to 11.68 log_2_ folds higher than that of control CEFs, which was over the entire time span significantly (*p* < 0.01) higher than Aseel CEFs (Figure 4c). Similarly, IFN-γ expresses strongly from the beginning and reaches its maximum at 13.9 log_2_ fold at 48 hpi. The expression levels of chIFITM1 and 3 were found to be higher than the other chIFITMs in Aseel and Kadaknath, respectively. 

## 4. Discussion

To measure the level of chIFITM gene expression against NDV in the current work, we employed the velogenic genotype XIII of the NDV strain that was used to infect CEF cells. The ND viral load increases, prompting the chicken embryo fibroblast (CEFs) cells to express significantly more chIFN-γ [28]. In Aseel’s CEF cells, there was noticeable upregulation starting at 6 hpi and continuing until 48 hpi. In contrast, in Kadaknath CEF cells, chIFN-γ expression began to increase significantly at 3 hpi and peaked (13.9 ± 0.49 log_2_ fold).

Further, it was significantly (*p* < 0.001) several folds higher than Aseel. However, interferons (IFNs), a vital component of innate immune signaling, serve as the first line of defense against invading viruses [29]. Therefore, it consistently and strongly expressed its opposition to the ND virus [30]. *chIFITM*s and *Mx* are members of the interferon-stimulating gene (*ISG*) group (mycovirus-resistant gene) [31,32]. Chicken has five members of the *IFITM* family: *chIFITM*1, *chIFITM*2, *chIFITM*3, *chIFITM*5, and *chIFITM*10. These protein genes are activated and made to express themselves by type I and *type II IFNs*, signifying the start of the innate host response in a negative feedback manner [14,33,34]. Recent studies confirm that *STAT/IRF* signaling pathways activate *IFITM* gene expression together with other *ISGs* during infection and inflammation [35]. This study examines the relative mRNA expression profile of chicken *IFITM*s after Newcastle disease virus (NDV) infection in vitro, with a focus on how the cells react during the early stages of NDV infection.

We observed significant upregulation of *chIFITM*s, *chIFN-γ*, and *Mx* in the CEF cells of both breeds. A significant viral load indicated the presence of a replicating virus in the CEF cells. *chIFITM*1, 2, 3, and 5 are noticeably and gradually upregulated in both breeds of CEFs after NDV infection. In Kadaknath CEF cells, *chIFITM*s and *IFN-γ* expressions were relatively high with statistical significance (*p* < 0.001) from 3 to 48 h post-infection compared to control uninfected cells. Researchers also found that high quantities of *chIFITM*1, 2, and 3 are expressed in CEFs from 4 to 24 h after H9N2 infection [36]. While CEF cells from Aseel took longer to exhibit strong mRNA expression of the *chIFN-γ* and *chIFITM* genes, they were strongly expressed at 6 h post-infection. In Kadaknath CEF, among the *IR-IFITM* family members, *chIFITM*3 (11.68 log_2_ folds) has the greatest expression observed at 6 h post-infection, followed by *chIFITM*2 (8.89 log_2_ folds) at 48 h post-infection, *chIFITM*1 (7.79 log_2_ folds), and *chIFITM*5 (7.04 log_2_ folds) at 24 h post-infection [14,16]. However, in Aseel, *IFITM*1 (6.00 log_2_ folds) came first, then *chIFITM*3 (3.46 log_2_ folds), then *chIFITM*2 (3.25 log_2_ folds) at 6hpi, and lastly *chIFITM*5 (3.82 log_2_ folds) at 24 hpi. Our finding that Kadaknath expressed large levels of *IFN-γ* and stimulated high levels of *chIFITMs* compared to Aseel is supported by studies from other scientists showing that *IFN*-treated CEFs expressed high levels of *chIFITMs* [14,33,36]. It is known that *chIFITM*1, 2, and 3 prevent the replication of a variety of RNA viruses that enter the host cell through endocytosis [9]. Infected CEF of Kadaknath showed a significant (*p* < 0.01) and robust overexpression of all *chIFITM*s starting at 3hpi when compared to the control. Whereas similarly in Aseel *chIFITM* genes, upregulation starts at 3 h post-infection compared to the control. A Significant and high-level *Mx* gene response has been observed at 3 h post-infection (4.18 log_2_ folds) and then a decreased level of expression similar to H9N2 infected CEFs [36], and it was delayed in Kadaknath at 24 h post-infection (3.57 log_2_ folds) [35]. The *Mx* gene was used as a positive control gene because it is one of the well-known *IFN*-stimulating genes and is highly expressed as a restriction factor in influenza A viral infection [37].

Results from other publications’ findings support in a similar way that the expression of *chIFITM*-2, 3, and *Mx* significantly increased after H3N8 infection, and this increase started at 6 h after infection. Although there was a reduction in *Mx* expression at 12 h after infection and both *chIFITM*2 and 3 were significantly elevated. At 6 h after infection, *chIFITM*1, 2, 3, and 5 and *Mx* expression significantly increased and persisted for 24 h in H5N3-infected CEF cells [36]. Like this, *IFITM*s were constantly and significantly upregulated in Kadaknath CEF cells throughout the study, which may be related to the increased level of *IFN-γ* gene expression. In contrast, Aseel CEFs expressed low levels of the *IFN-γ* and *chIFITM* genes investigated. This is supported by the findings of other papers. Thus, it was hypothesized that interferons would activate and upregulate the expression of *chIFITM* in CEFs based on evidence from researchers Whitehead and Smith et al. [15,36]. It was further demonstrated that it is possible to inhibit virus replication by simply preventing access to a cell, as evidenced by the production of *IFITM*s following *IFN* treatment [29,33].

We measured the log_10_ viral copies of NDV at 3-, 6-, 12-, 24-, and 48-h post-infection in CEF cells and contrasted both breeds. The viral load steadily increased from 3 h post-infection itself in both breeds, and the viral load was significantly (*p* < 0.01) lower in Kadaknath when compared to Aseel from 3 to 12 h post-infection and overall load through the infected period. The outcomes showed that the ND viral load at 12 h post-infection in Kadaknath CEF (2.15 log_10_) cells was reduced by 60% in comparison to Aseel CEF (5.38 log_10_) cells. Inversely proportional to viral load, Kadaknath showed significantly (*p* < 0.01) higher expression of all the *chIFITM* genes at all periods than Aseel. The significant upregulation of chIFITM3 has been observed to effectively reduce the infectivity of the Newcastle disease (ND) virus and inhibit its multiplication in CEF cells [38]. Like this, Blyth et al. found that overexpressing *chIFITM*3 reduces influenza H6N2 and H1N9 strain infection in DF-1 cells by 30 to 40% [39]. Similarly to this, in vitro overexpression of *chIFITM*3 limits the multiplication of the influenza virus by 55% [15]. Infectious bursal viral (IBV) strains of QX, M41-CK, and Beaudette infection significantly upregulate all *IR-chIFITM* genes at 24 hpi [14]. Scientists concluded that *chIFITM*2 and 3 greatly decreased the lyssavirus infection [14,15].

However, by altering the characteristics of cellular membranes and blocking the cell surface receptors to restrict viral entry, the Interferon-inducible transmembrane proteins (*IFITM*s) prevent many harmful viruses from infecting cells and causing infection [40,41,42]. This ultimately prevents viral fusion [41]. Several reports confirm that the *IFITM*s effectively control RNA viruses such as avian influenza A virus (IAV), lyssaviruses [15], infectious bronchitis virus (IBV) [14], and avian reovirus multiplication (ARV) [33], which follow the endosomal pathway to enter the host cell membrane for multiplication [43].

Microscopic analysis of the infected cells showed that the monolayer remained intact, much like in mock-infected cells, and that no CPE occurred until 3 h post-infection (Aseel) or 6 h post-infection (Kadaknath). However, after 6 to 12 hpi, the light-microscopy analysis revealed morphological changes indicative of CPE, including rounding, the fusing of infected cells to form syncytia, and the detachment of cells from the monolayer followed by cell death. Cellular rounding, membrane blebbing, cytoplasm vacuolation, nuclear condensation, and nuclear envelope collapse are among the morphological changes brought on by NDV infection. This result is congruent with what has been documented in other publications [44]. The CPE in the current investigation, however, showed that CEF cells of both breeds began exhibiting CPE sooner than had been previously reported. According to Li et al., overexpression of *IFITM*3 inhibited the inflammatory response of PF15 cells and is crucial to the *TLR4*-*NF*-*B* signaling pathway, which is implicated in the inflammatory response [45]. In Kadaknath, *chIFITM*3 expression levels are consistently high (*p* < 0.01), with a log_2_ fold ranged 10.08 to 11.68 in contrast to 1.00 to 3.46 in Aseel. As a result, compared to Aseel, the Kadaknath CEF cells had a delayed cytopathic effect and cell death. Elevated *IFITM* gene expression inhibits the spread of infections by restricting host cell proliferation. It is also involved in inhibiting cell adhesion and controlling cell growth [46]. Additionally, Anjum et al. noticed a decrease in cytopathic effects in *chIFN*-treated CEF cells when they were infected with ND and AIV [47]. Similarly, in Kadaknath, CEF cells expressed a high level of *IFN-γ,*and delayed cytopathic changes were observed.

## 5. Conclusions

In this study, we have shown the variability in the magnitude of *chIFITM*s mRNA expression between breeds during Newcastle disease Viral infection. Such variation suggests that the *chIFITM* response may be breed-dependent and intragenic factor-dependent. Our data suggest CEF cells start expressing all *chIFITM* genes significantly in the early stage of infection, regardless of the breed of the chicken. Elevated levels of expression of *chIFITM*s in Kadaknath CEF cells restrict viral multiplication compared to the Aseel CEFs. Together, the results show *chIFITM*s play a critical role in restricting the ND virus’s multiplication. In addition, it has been shown that the basal level of *IFN-γ* expression will impact *chIFITM* gene expression. Therefore, we have revealed that viral entry is restricted depending on the level of *chIFITM* expression, and the expression depends on other factors. This study was conducted in vitro, and more experiments are necessary to clarify the underlying mechanism for controlling viral diseases in chickens. In the future, in ovo and in vivo studies will be required to better understand the role of this gene in the immune system.

## Figures and Tables

**Figure 1 biology-12-00919-f001:**
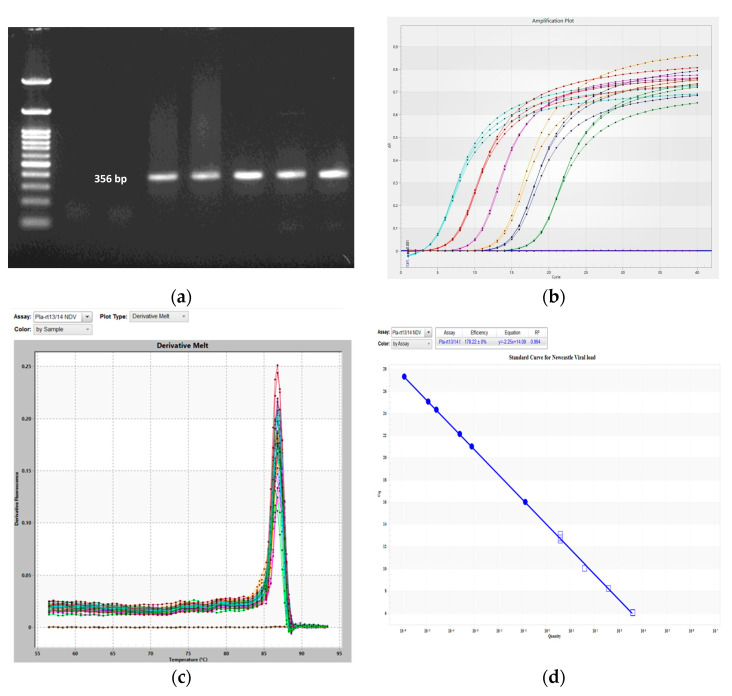
Newcastle disease virus detection and quantification. (**a**) PCR amplification of a 356 bp−long *F*−*gene* amplicon specific to the ND virus. (**b**–**d**) Real−time quantitative PCR assay creation for absolute quantification. (**b**) Amplification plot of serially diluted T−NP plasmid standard (each color indicates the different concentration T-NP plasmid at 10-1 to 10-6 dilution). (**c**) Melting curve. (**d**) Standard curve for ND viral load detection with linear regression equation *Y* = −2.25x + 14.09 and R^2^ score value 0.994.

**Figure 2 biology-12-00919-f002:**
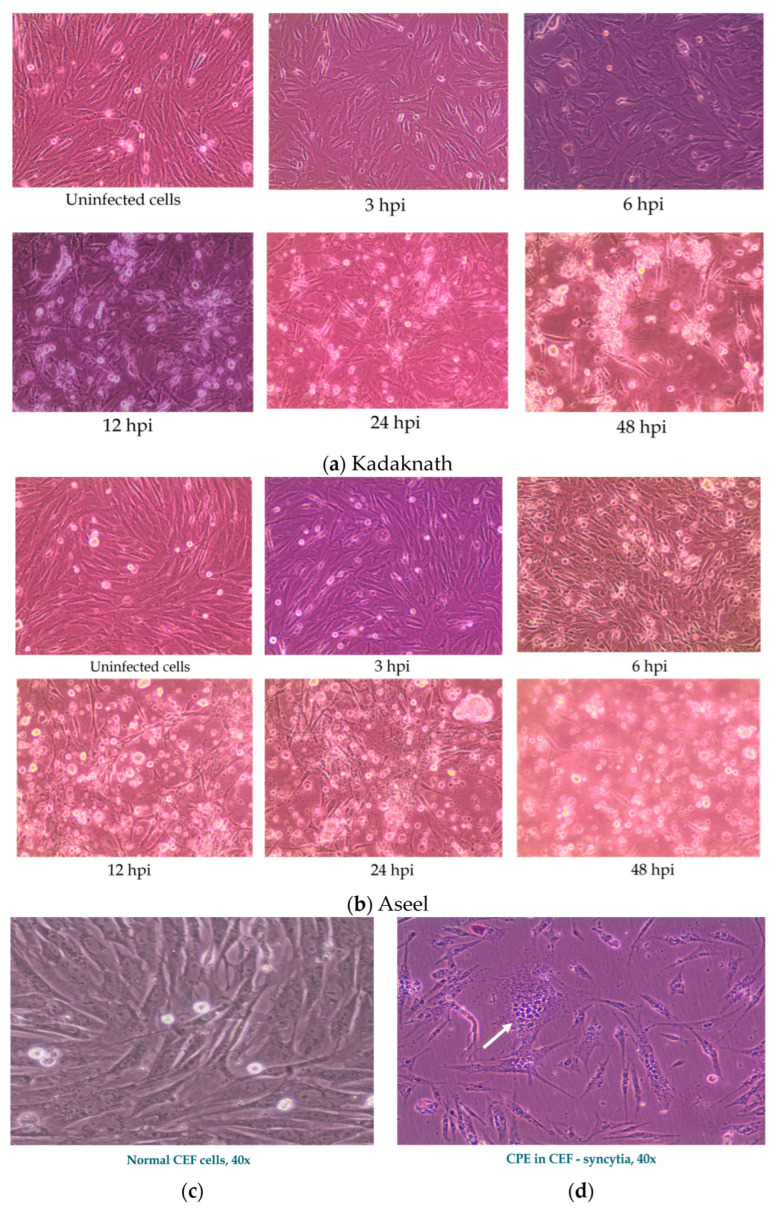
Observations of Chicken embryo fibroblast cell cultures under an inverted microscope exhibiting at 20× (**a**,**b**) and 40× (**c**,**d**) phase contrast objectives. (**a**,**b**). Uninfected and Newcastle disease virus-infected CEF cells at different hours post-infection (hpi). (**a**) Kadaknath CEF showed delayed cytopathic changes, with cell rounding noticed at 12 h, and (**b**) Aseel CEF started showing cytopathic changes, such as cell rounding, at 6 h. (**c**) Normal CEF cells and (**d**) cytopathic effect of ND virus in CEF showing multinucleated cells (arrow indicates formation of Syncytia).

**Figure 3 biology-12-00919-f003:**
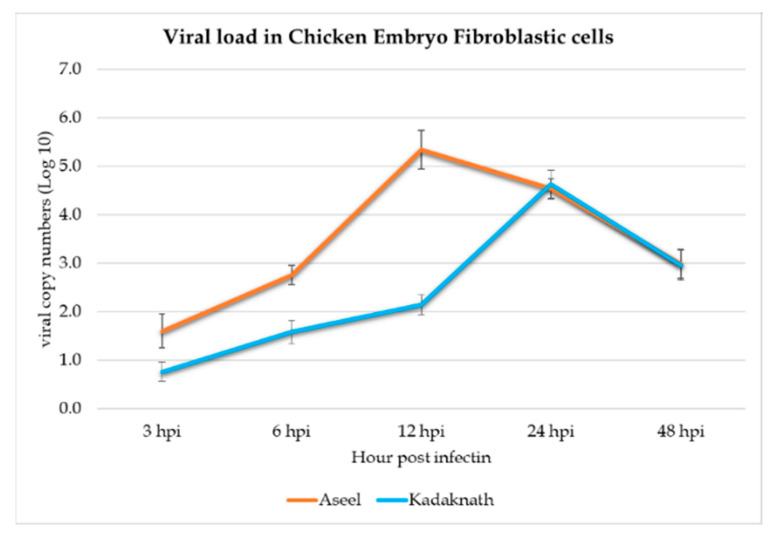
A line graph depicts the absolute quantity of viral load based on T-NP plasmid standards at various time points in CEF cells of Aseel and Kadkanth.

**Figure 4 biology-12-00919-f004:**
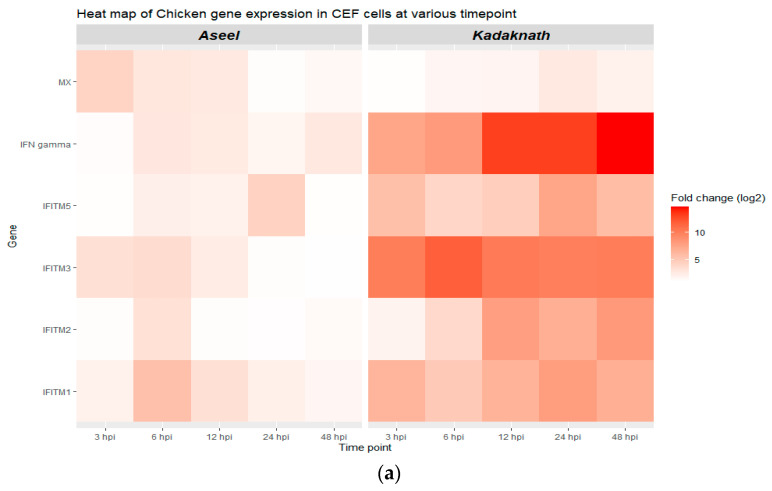
Gene expression analysis in CEF of Aseel and Kadaknath. (**a**,**b**) Heatmap and boxplot, respectively, illustrating the various levels of gene expression at different hours post-infection (hpi) of ND virus infection. (**c**) Boxplot showing the cumulative gene expression of each gene in the Kadaknath and Aseel throughout time.

**Table 1 biology-12-00919-t001:** List of primer sequences for qRT-PCR.

Gene	Primer	Sequence (5′->3′)	Reference
*chIFITM*1	FP	GCAGGATGTGACCACCACTA	NM_001350059.2
	RP	CTTCGCTGTCCTCCCATAGC	
*chIFITM*2	FP	AACAGGCGGAGGTGAGCAT	NM_001350058.2
	RP	AAGATGAGCGAGGGGAAGCA	
*chIFITM*3	FP	CGTGAAGTCCAGGGATCGCA	NM_001350061.2
	RP	GCAACCAGGGCGATGATGAG	
*chIFITM*5	FP	CCAACCCCACTTCTGGACGA	NM_001199498.1
	RP	ATCACTCCGAAGGGCACGAC	
*chMx*	FP	GTCCAAGAGGCTGAATAACAGAG	NM_204609
	RP	GTCGGATCTTTCTGTCATATTGG	
*chIFN-γ*	FP	TGAGCCAGATTGTTTCGATG	[27]
	RP	CTTGGCCAGGTCCATGATA	
*chβ-Actin*	FP	TATGTGCAAGGCCGGTTTC	
	RP	TGTCTTTCTGGCCCATACCAA	
*NDV-NP*	Pla-rt13	CAACAATAGGAGTGGAGTGTCTGA	[25]
	Pla-rt14	CAGGGTATCGGTGATGTCTTCT	

**Table 2 biology-12-00919-t002:** Chicken embryo fibroblastic cells’ gene expression fold changes (log_2_ (2^−ΔΔCt^)) in response to the Newcastle disease virus infection.

Breed	Time Point	*IFITM*1	*IFITM*2	*IFITM*3	*IFITM*5	*IFN*-γ	*MX*
	3 hpi	2.64 ± 0.40 ^c^	1.06 ± 0.01 ^c^	2.87 ± 0.26 ^b^	1.01 ± 0.00 ^b^	1.17 ± 0.09 ^c^	4.18 ± 0.38 ^a^
	6 hpi	6.00 ± 0.25 ^a^	3.25 ± 0.24 ^a^	3.46 ± 0.42 ^a^	2.07 ± 0.16 ^b^	2.28 ± 0.24 ^ab^	3.35 ± 0.33 ^b^
Aseel	12 hpi	3.88 ± 0.35 ^b^	1.06 ± 0.00 ^c^	2.38 ± 0.71 ^b^	2.41 ± 0.32 ^b^	2.18 ± 0.18 ^b^	3.34 ± 0.38 ^b^
	24 hpi	2.19 ± 0.09 ^c^	1.22 ± 0.12 ^bc^	1.26 ± 0.16 ^c^	3.82 ± 0.30 ^a^	2.13 ± 0.28 ^b^	1.03 ± 0.01 ^c^
	48 hpi	1.91 ± 0.20 ^cd^	1.51 ± 0.08 ^b^	1.00 ± 0.00 ^c^	1.53 ± 0.27 ^c^	2.59 ± 0.34 ^a^	1.43 ± 0.06 ^c^
	3 hpi	6.33 ± 0.24 ^b^	2.39 ± 0.38 ^d^	10.44 ± 0.26 ^b^	5.63 ± 0.17 ^b^	7.46 ± 0.45 ^c^	1.01 ± 0.05 ^c^
	6 hpi	4.77 ± 0.38 ^c^	3.25 ± 0.29 ^c^	11.68 ± 0.36 ^a^	3.18 ± 0.46 ^c^	7.77 ± 0.21 ^c^	1.38 ± 0.16 ^c^
Kadaknath	12 hpi	6.77 ± 0.21 ^b^	7.02 ± 0.47 ^b^	10.08 ± 0.10 ^b^	4.28 ± 0.19 ^bc^	12.38 ± 0.54 ^b^	1.66 ± 0.10 ^c^
	24 hpi	7.79 ± 0.52 ^a^	7.48 ± 0.36 ^b^	10.47 ± 0.36 ^ab^	7.04 ± 0.27 ^a^	12.56 ± 0.39 ^b^	3.57 ± 0.56 ^a^
	48 hpi	7.48 ± 0.36 ^a^	8.89 ± 0.33 ^a^	11.00 ± 0.51 ^a^	6.22 ± 0.27 ^a^	13.9 ± 0.49 ^a^	2.56 ± 0.46 ^b^

Mean value with different superscript shows significant difference at *p* < 0.05.

## Data Availability

The data set created and analyzed in the current study will be made available upon reasonable request.

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
