# Peer review of "In Vitro Characterization of chIFITMs of Aseel and Kadaknath Chicken Breeds against Newcastle Disease Virus Infection"

_biology, 2023, doi:10.3390/biology12070919_

Round 1
Reviewer 1 Report
In the manuscript " In-Vitro characterization of chIFITMs of Aseel and Kadaknath 2 chicken breeds against Newcastle disease virus infection" the authors have characterised chIFITMs of Aseel and Kadaknath chicken breeds against Newcastle disease virus infection. Kadaknath is known for its resistance to ND, and the current manuscript authors are trying to address it with interesting findings. But some more critical corrections are needed before accepting the manuscript.
Major point:
- The study lacks control, and It Should include the CEF of a susceptible chicken or DF1 cell line as a standard control.
- The authors believe that 'CEF cells start expressing all chIFITM genes significantly in the early stage of infection, regardless of the breed of the chicken’ What about IFN pretreatment and fold change I believe that IFN vs NDV in both dose-dependent manner would address their claims.
- Is there any sequence difference in chIFITMs between these breeds?
- Microscopy images are very poor quality; they should be replaced. Scale is missing in all the images. Authors should use a better microscope with phase contrast.
Some minor points
- In simple summary, the authors should add more points about the results. The sentence ‘The genes Chicken Interferon Inducible Transmembrane Proteins (chIFITM), which produces chicken interferon-inducible transmembrane proteins,’ is a bit repetitive. Kindly rephrase the sentence.
- Formatting errors, minor English editions
minor English editions
Author Response
Dear Sir
Thank you for reviewing and i have enclosed the reply for the comments for your kind perusal
kind regards
a.k.thiruvenkadan

Reviewer 2 Report
The authors conducted an important study on the immune response of fibroblasts in two chicken breeds. The staging of the experiment is logical and reasonable. However, it is not clear how many embryos were taken for the experiment? The genetic basis of the variability of the studied genes is of interest. Were there such studies on these breeds? It would be desirable to add such information to the discussion or to the results, as conclusions were made about the genetic resistance to diseases of the Kadaknath breed. After making corrections, the article will be ready for publication.
Author Response
Dear sir
Thank your for the comments and I have corrected the same reply for your kind perusal
kr
a.k.thiruvenkadan

Reviewer 3 Report
The paper, “In-Vitro characterization of chIFITMs of Aseel and Kadaknath chicken breeds against Newcastle disease virus infection” by Muthusamy Malarmathi et al. analyzed the level of chIFITM, MX and IFN-γ gene expression against the Newcastle disease virus (NDV) in chicken embryo fibroblast cells of Aseel and Kadaknath 2 chicken. The experimental design and data are clear, but the innovation is insufficient.
Major issues:
The relationship between chIFITM and NDV has been documented in a few papers(PMID: 33762417, PMID: 30566670, PMID: 28435159).
In Abstract, the authors mentioned that “The result suggests that the Kadaknath chicken breed may have a higher level of disease tolerance compared to the Aseel chicken. ”. The data presented in this study was obtained at the cellular level using CEF cells. However, immune responses in an organism are complex processes. Therefore, to draw this conclusion, it is recommended to conduct animal experiments.
Minor issues:
1. Figure 3 need a higher resolution version.
2. Line 72: “influenza virus and its titre in in vitro study” should be “influenza virus and its titer in vitro study”.
3. Line 334: “because it is a one of” should be “because it is one of”.
4. Line 61-62: Please double check the exchange rate between the US dollar and the Indian rupee.
Minor editing of English language required.
Author Response
Dear sir
thank you for the comments and I have enclosed the reply for your kind perusal
a.k.thiruvenkadan

Round 2
Reviewer 1 Report
I would appreciate the authors for the present work. The Kadaknath chicken breed is known for its resistance to viral infection, the present study shows a possible mechanism.
Reviewer 3 Report
The revision looks good.